# Development of Hydrophobic Cell-Penetrating Stapled Peptides as Drug Carriers

**DOI:** 10.3390/ijms241411768

**Published:** 2023-07-21

**Authors:** Keisuke Tsuchiya, Kanako Horikoshi, Minami Fujita, Motoharu Hirano, Maho Miyamoto, Hidetomo Yokoo, Yosuke Demizu

**Affiliations:** 1Division of Organic Chemistry, National Institute of Health Sciences, Kawasaki-shi 210-9501, Japan; k.tsuchiya@rs.socu.ac.jp (K.T.); s202095f@yokohama-cu.ac.jp (K.H.); w225425e@yokohama-cu.ac.jp (M.F.); w225510f@yokohama-cu.ac.jp (M.H.); s202107d@yokohama-cu.ac.jp (M.M.); 2Division of Pharmaceutical Organic Chemistry, Faculty of Pharmaceutical Sciences, Sanyo-Onoda City University, 1-1-1 Daigakudori, Sanyo-Onoda-shi 756-0884, Japan; 3Graduate School of Medical Life Science, Yokohama City University, 1-7-29 Suehiro-cho, Tsurumi-ku, Yokohama-shi 230-0045, Japan; 4Graduate School of Medicine, Dentistry and Pharmaceutical Sciences, Okayama University, 1-1-1 Tsushimanaka, Kita-ku, Okayama-shi 700-8530, Japan

**Keywords:** cell-penetrating peptide, stapled peptide, hydrophobic peptide, helical structure, plasmid DNA delivery

## Abstract

Cell-penetrating peptides (CPPs) are widely used for the intracellular delivery of a variety of cargo molecules, including small molecules, peptides, nucleic acids, and proteins. Many cationic and amphiphilic CPPs have been developed; however, there have been few reports regarding hydrophobic CPPs. Herein, we have developed stapled hydrophobic CPPs based on the hydrophobic CPP, TP10, by introducing an aliphatic carbon side chain on the hydrophobic face of TP10. This side chain maintained the hydrophobicity of TP10 and enhanced the helicity and cell penetrating efficiency. We evaluated the preferred secondary structures, and the ability to deliver 5(6)-carboxyfluorescein (CF) as a model small molecule and plasmid DNA (pDNA) as a model nucleotide. The stapled peptide **F-3** with CF, in which the stapling structure was introduced at Gly residues, formed a stable α-helical structure and the highest cell-membrane permeability via an endocytosis process. Meanwhile, peptide **F-4** demonstrated remarkable stability when forming a complex with pDNA, making it the optimal choice for the efficient intracellular delivery of pDNA. The results showed that stapled hydrophobic CPPs were able to deliver small molecules and pDNA into cells, and that different stapling positions in hydrophobic CPPs can control the efficiency of the cargo delivery.

## 1. Introduction

Cell-penetrating peptides (CPPs) are widely used for the intracellular delivery of small molecules, peptides, nucleic acids, and proteins [1]. CPPs can be classified as cationic, amphipathic, and hydrophobic based on the physicochemical properties of the sequence. Many cationic and amphipathic peptides have been developed for the intracellular delivery of various cargo molecules. However, there have been comparatively few reports on hydrophobic CPPs [2]. The development of a drug delivery system (DDS) using hydrophobic CPPs is considered difficult because most hydrophobic CPPs have low solubility in aqueous solutions and aggregate easily [3]. However, hydrophobic CPPs are expected to reduce the risk of cytotoxicity and have less accumulation in organs compared with cationic CPPs [4]. TP10 is a hydrophobic CPP, which has four cationic amino acid (Lys) residues among many hydrophobic amino acid residues, and forms a helical structure [5,6]. TP10 is a chemically synthesized transportan [6,7] analog, derived from the *N*-terminal domain of the neuropeptide galanin linked through a lysine residue to mastoparan [8]. TP10 can cross the cell membrane and has demonstrated the transport of several cargoes, including drugs. The mechanism of the entry of these transporter–cargo complexes into the cells was mostly dependent on endocytosis [9].

Cationic cell-penetrating peptides (CPPs) interact electrostatically with the anionic phosphate groups of DNA and RNA, forming polyion complexes [1,10]. These complexes can form nanoparticles with suitable sizes and zeta potentials, facilitating their internalization into cells through endocytosis. The specific sequence of CPPs can modulate the physicochemical properties of the peptide/nucleic acid complexes and their efficiency in delivering genetic material into cells [11,12,13]. In our previous work, we reported on the use of cationic CPPs composed of arginine (Arg) and α,α-disubstituted amino acid (dAA) residues as tools for delivering plasmid DNA (pDNA) [14]. By incorporating dAAs into CPPs, we were able to induce a stable helical structure and increase resistance to protease digestion, leading to enhanced and prolonged cell-penetrating abilities.

Stabilizing the secondary structures of CPPs, especially the helical structures, is an efficient strategy to increase the cell membrane permeability and deliver cargo molecules [14]. A useful tool for stabilizing the α-helical structures of peptides is the side chain stapling of amino acids to form stapled peptides [15]. Such stapled peptides have been synthesized using ring-closing metathesis between two (*S*)-2-(4-pentenyl)-alanine (S_5_) residues at the *i* and *i* + 4 positions to form the hydrocarbon staple [16]. Covalent cross-linking at the *i* and *i* + 7 positions has also been performed between S_5_ residues and (*R*)-2-(7-octenyl)-alanine residues. In addition, perfluoroarene cross-linked TP10, which could efficiently permeate the blood–brain barrier, has been synthesized, in which the structure of TP10 was optimized by stapling the side chains to improve the cell membrane permeability [17]. However, few studies have applied the stapling of side chains to control the helical structures of hydrophobic CPPs for use as carriers in DDSs. The modification of hydrophobic CPPs is needed to optimize the properties of the hydrophobic CPPs and to develop effective DDS techniques. In the present study, we examined the most common stapling structure, i.e., the stapling of aliphatic carbon side chains, to control the helical structure of TP10 while maintaining the hydrophobicity of the peptide. Stapled hydrophobic CPPs were designed based on TP10, and the preferred secondary structures and ability to deliver 5(6)-carboxyfluorescein (CF) as a model small molecule and pDNA as a model nucleotide were evaluated (Figure 1). The physicochemical properties of peptide/pDNA complexes were characterized using zeta-potential measurements, dynamic light scattering (DLS) measurements, and gel retardation assay. In vitro experiments of transfection and cellular uptake using peptide/pDNA complexes were conducted with HEK293 cells. Our objective is to design hydrophobic CPPs incorporating unnatural amino acids, particularly those with a stapling structure, for efficient pDNA delivery.

## 2. Results

### 2.1. Design of Stapled TP10 Peptides

Focusing on the hydrophobic face of the helix of TP10 (**F-1**), a hydrophobic stapling structure was introduced into the sequence while maintaining the hydrophobicity of the peptide. First, to stabilize the α-helical structure of TP10, the stapled peptide **F-2** with two S_5_ residues replacing the Gly residues, which generally destabilize helix structures, was synthesized (Table 1). Next, the stapled peptides **F-3** and **F-4** were constructed by replacing two pairs of Leu and Ile residues near the *N*- and *C*-termini with S_5_ residues, which introduced stapling without appreciably changing the number of carbon atoms or the hydrophobicity of the peptide. To evaluate the ability of these TP10 derivatives to deliver small molecules, CF [18] was selected as a model small molecule and conjugated to the *N*-terminus of each peptide. A conventional Fmoc-based solid-phase method was performed to synthesize the designated peptides. Briefly, after the elongation of the sequence, stapling was carried out on resin using the ring-closing metathesis reaction for peptides **F-2**, **F-3**, and **F-4**. All peptides were cleaved from resin using a trifluoroacetic acid (TFA) cocktail and purified by reversed-phase high-performance liquid chromatography (RP-HPLC), and identified using IT-TOF MS.

### 2.2. Secondary Structural Analysis of Stapled TP10 Peptides

CD measurements were performed to analyze the preferred secondary structures of the synthesized peptides **F-1**–**F-4** (10% acetonitrile in 20 mM phosphate buffer, pH 7.4) (Figure 2). The parent peptide, **F-1**, exhibited a CD spectrum characterized by a positive Cotton effect at approximately 190 nm and negative Cotton effects at approximately 209 and 222 nm. These characteristic Cotton effects indicated the presence of a helical structure in **F-1**. Similarly, the stapled peptides **F-2** to **F-4** displayed CD spectra with similar patterns to that of **F-1**, suggesting the formation of helical structures in all of these peptides. To further evaluate the dominant secondary structures of the peptides, a parameter called the R value (θ_222_/θ_208_) was calculated. The *R* value represents the ratio of the CD signal intensities at 222 nm and 208 nm, respectively. For the peptides **F-1**, **F-2**, **F-3**, and **F-4**, the *R* values were determined to be 0.72, 1.83, 0.95, and 1.58, respectively. Based on previous studies [19,20], these *R* values indicate that the dominant secondary structures of all the peptides were α-helices. Among the peptides, **F-2** exhibited the highest *R* value, suggesting that the α-helical structure of **F-2** was the most strongly stabilized compared to the other peptides.

### 2.3. Evaluation of the Delivery of Small Molecules by Stapled TP10 Peptides

First, the delivery of CF, as a model small-molecule cargo, into HEK293 cells by the stapled TP10 peptides was evaluated using flow cytometry to detect the intracellular fluorescence intensity from CF, which indicated the amount of CF delivered by the peptides in the cells (Figure 3). The parent peptide, **F-1**, exhibited a decrease in intracellular fluorescence intensity over time. In contrast, the stapled peptides (**F-2**, **F-3**, and **F-4**) displayed similar intracellular fluorescence intensities at both 2 and 48 h post-treatment. Notably, the C-terminal stapled peptide, **F-3**, exhibited the highest intracellular fluorescence intensity at both time points. On the other hand, the other stapled peptides, **F-2** and **F-4**, demonstrated lower intracellular fluorescence intensities than the parent peptide, **F-1**, at 2 h but higher intensities compared to **F-1** at 48 h. These results suggest that the stapled peptides possess increased resistance to cellular degradation and maintained cellular uptake compared to the parent peptide, **F-1**. Moreover, the optimization of the stapling position in the peptides appears to enhance the efficiency of small molecule delivery. This suggests that specific modifications, such as the strategic placement of the staples, can improve the effectiveness of peptide-based delivery systems for small molecules.

To elucidate the mechanism of the delivery of small molecules by **F-3**, which showed the highest cellular uptake, and **F-1**, the parent peptide, we evaluated the cellular uptake after treatment with endocytosis inhibitors, and after complete inhibition of endocytosis at 4 °C. To inhibit specific endocytic pathways, three different inhibitors were used: amiloride, nystatin, and sucrose [21,22,23]. Each inhibitor targets a distinct endocytic pathway: amiloride inhibits micropinocytosis (a form of endocytosis where cells engulf fluid and solutes); nystatin inhibits caveolae-mediated endocytosis (a process involving small invaginations in the cell membrane called caveolae); and sucrose inhibits clathrin-mediated endocytosis (a mechanism in which the protein clathrin forms coated pits on the cell surface to internalize molecules). As a control, a peptide called nona-arginine (R9) with CF (chloroform) was used. This peptide is known to enter cells via endocytosis. The uptake of **CF-R9** with that of the parent peptide **F-1** and the stapled peptide **F-3** was compared. The results indicated that the uptake of both the parent peptide **F-1** and the stapled peptide **F-3** was significantly inhibited when treated with the endocytosis inhibitors and when incubated at 4 °C. This suggests that the TP10 derivatives (**F-1** and **F-3**) enter the cells through an endocytic process similar to that observed for **CF-R9** (Figure 4a,b).

### 2.4. pDNA Delivery by Stapled TP10 Peptides

The ability of peptides **F-1**–**F-4** to deliver pDNA was evaluated using complexes between the peptides and pDNA in 10 mM Hepes buffer (pH 7.3). The zeta potentials and sizes of the peptide/pDNA complexes prepared at various charge ratios were determined (Table 2). The size of the **F-1** peptide/pDNA complexes at a charge ratio of 2 was 3364 nm, and the sizes decreased with an increase in the charge ratios (Table 2). The sizes of the peptide/pDNA complexes were in the order of several hundred nm, which is considered to be a useful size for the delivery of pDNA, except for the complex with **F-1** at N/P = 2, which was defined as the residual molar ratio of the amino groups in polycations of peptides (N) to the phosphate groups in pDNA (P). The complexes with **F-1**–**F-4** showed similar zeta-potentials that reached >+15 mV. Comparing the characteristics of the complexes formed with pDNA at N/P = 2, 4, and 8, all the peptides exhibited similar particle sizes and zeta-potentials, at a charge ratio of 8. Thus, we examined the complexes of **F-1**–**F-4** at N/P = 8 to evaluate the effect of the stapling of TP10 on the ability to deliver pDNA.

The intracellular uptake of Cy5-labeled pDNA complexed with peptides **F-1**–**F-4** was evaluated in HEK249 cells. The fluorescence intensity of Cy5 incorporated into HEK249 cells was quantified using flow cytometry at 24 h after administration (Figure 5). The complexes with **F-2** and **F-3** showed similar fluorescent intensity from Cy5 to that of the parent peptide **F-1**. However, the intracellular intensity of Cy5 after transfection with **F-4** was significantly higher than that with **F-1**–**F-3**. When the peptide **F-1**–**F-4**/Cy5-pDNA complex was added, the strongest fluorescence from peptide detected by CF was observed in the case of **F-3**, which showed a good ability to permeate cell membranes (Appendix A). The stability of the peptide/pDNA complex was then evaluated by adding the negatively charged competitive inhibitor, dextran sulfate. The **F-3**/pDNA complex dissociated at the lowest concentration of dextran sulfate, 0.5 mg/mL, suggesting that the **F-3**/pDNA complex was less stable than the other complexes (Appendix A). These results suggested that **F-4** may have remained in the medium and contributed to the maintenance of the complex, which increased the efficiency of the pDNA delivery, while **F-3** detached from the pDNA and may have been able to migrate into the cells. These results suggested that the efficiency of pDNA delivery depended not only on the presence or absence of the stapling structure and α-helicity of the peptide, but also on the position of the stapling.

## 3. Discussion

Focusing on the secondary structure and hydrophobicity of TP10 **F-1**, hydrophobic stapling was introduced into **F-1** to construct three stapled peptides **F-2**–**F-4** and the functionality of the peptides was evaluated. Secondary structure analysis of **F-1**–**F-4** revealed that all the peptides formed α-helical structures. In particular, the stapled peptide **F-2**, in which the stapling was introduced between Gly residues, formed the most stable α-helix. Then, the ability of **F-1**–**F-4** to deliver small molecules was evaluated, and the stapled peptide **F-3** showed the highest cellular uptake after both 2 and 48 h. The uptake pathways of the peptides in the presence of endocytosis inhibitors and at 4 °C they were also examined, and it was found that the uptake was clearly inhibited, indicating that the intracellular uptake was via endocytosis. The efficiency of the delivery of Cy5-labeled pDNA by the TP10 derivatives was evaluated and the highest intracellular uptake was observed for **F-4**. However, **F-3** could more efficiently deliver a small molecule cargo, compared with **F-4**. When the peptide/Cy5-pDNA complex was added, higher fluorescence of peptide, that was detected using CF, was observed with **F-3** than with **F-4**, which suggested that **F-3** detached from the pDNA and migrated into the cells, while **F-4** remained in the medium and contributed to maintaining the structure of the complex. The TP10 derivatives developed in this study are promising peptides as delivery tools for small molecules and nucleic acids. The introduction of different stapling structures showed different effects on the delivery ability, suggesting that controlling the stapling position in hydrophobic CPPs can affect the delivery of different cargoes. For the development of DDS carriers using hydrophobic CPPs, controlling the helical structure by stapling aliphatic carbon side chains and optimizing the properties of hydrophobic CPPs can be an effective DDS methodology. In addition to the hydrocarbon staples formed at the *i* and *i* + 4 positions between the two S5 residues used in this article, aliphatic covalent bridges at the *i* and *i* + 7 positions between the common S5 and (R)-2-(7-octenyl)-alanine residues may also be useful as modifications suitable for optimizing hydrophobic CPP. This methodology is expected to be useful in the development of hydrophobic CPPs as delivery tools.

## 4. Materials and Methods

All of the coupling reagents were obtained from Tokyo Chemical Industry Co., Ltd. (Tokyo, Japan), and were used as supplied without further purification. Fmoc-protected amino acids were obtained from Tokyo Chemical Industry Co., Ltd. (Tokyo, Japan) and Watanabe Chemical Industries, Ltd. (Hiroshima, Japan). They were used as received unless otherwise noted.

### 4.1. Peptide Synthesis

The peptides were synthesized using microwave-assisted Fmoc-based solid-phase methods on a Liberty Blue™ Automated Microwave Peptide Synthesizer (CEM corp. (Tokyo, Japan)) (Appendix A). A representative coupling and deprotection cycle are described as follows. Rink Amide Protide (LL) resin was soaked in CH_2_Cl_2_. After the resin had been washed with DMF, Fmoc-amino acid (5 equiv.), Oxyma Pure (10 equiv.), and DIC (10 equiv.) dissolved in a solution of DMF were added to the resin. For additional coupling, the same conditions, or Fmoc-amino acid (6 equiv.), Oxyma Pure (6 equiv.) and DIC (6 equiv.) dissolved in a solution of DMF, were added to the resin. Fmoc protective groups were deprotected using 20% piperidine in DMF. Ring closing metathesis reactions were performed using 20 mol% 2nd generation Grubbs catalyst in 1,2-dichloroethane for 15 min at 60 °C by MARS6 (CEM corp.). This cycle was performed two to four times. The peptide was suspended in cleavage cocktail (95% TFA, 2.5% water, 2.5% triisopropylsilane) at 42 °C for 30 min on Razor (CEM corp.) to cleave from the resin. TFA was evaporated to a small volume under a stream of N_2_ and dripped into cold ether to precipitate the peptide. The crude peptide was purified by using a JASCO preparative HPLC system equipped with Inertsil WP300 C18 column (20 × 250 mm, 5 µm, GL Science (Tokyo, Japan)) or XBridge^Ⓡ^ Prep C18 OBD (19 × 250 mm, 5 µm, Waters (Shinagawa City, Japan)). Mobile phases A and B were 0.1% TFA in water and 0.1% TFA in MeCN, respectively. The fractions containing the target product were collected, and lyophilized to obtain the desired peptide. The purified peptides were analyzed by a JASCO analytical HPLC system equipped with an InertSustainSwift C18 column (4.6 × 250 mm, 3 µm HP, GL Sciences) or Inertsil WP300 C18 column (4.6 × 250 mm, 5 µm, GL Sciences). Mobile phases A and B were 0.1% TFA in water and 0.1% TFA in MeCN, respectively. MS identification was performed by a Shimazu IT-TOF MS equipped with an electrospray ionization source. The analytical data for the synthesized peptides are provided in Supporting Information.

### 4.2. Circular Dichroism Spectroscopy

CD spectra were recorded with a JASCO J-1100 CD spectrometer using a 1.0 mm path length cell. The data are expressed in terms of [θ]; i.e., total molar ellipticity (deg cm^2^ dmol^−1^). Peptides were dissolved in 10% MeCN in 20 mM phosphate buffer solution at 50 µM concentrations. Each parameter was: wavelength: 190–260 nm; bandwidth: 1 nm; response: 1 s; speed: 50 nm min^−1^; accumulations: 3 times; and path length: 1 nm.

### 4.3. Cell Culture

HEK293 cells were cultured in Dulbecco’s modified Eagle’s medium (DMEM) (Sigma–Aldrich, St. Louis, MO, USA). The medium was supplemented with 10% fetal bovine serum (FBS) (Sigma–Aldrich), 100 U/mL Penicillin (Nacalai Tesque, Kyoto, Japan), 100 µg/mL streptomycin (Nacalai Tesque, Kyoto, Japan). All cells were incubated at 37 °C in a humidified atmosphere of 5% CO_2_ in air.

### 4.4. Flow Cytometry

HEK293 cells were seeded in 24-well plates at a density of 200,000 cells/well and cultured in DMEM for 21 h. Then, the cells were treated with the relevant peptide (peptide concentration; 5 µM) and incubated for the appropriate time. After 2 and 48 h, the cells were washed with PBS containing heparin (20 units/mL) and added again, and detached by pipetting. The collected cells were washed once with PBS. Then, the cells were suspended in 500 mL of PBS and then analyzed by a BD accuri C6 Plus flow cytometer (BD Bioscience, Franklin Lakes, NJ, USA).

### 4.5. Inhibition of Endocytosis

HEK293 cells were seeded in 24-well plates at a density of 200,000 cells/well and cultured in DMEM for 23 h. After the medium had been replaced with fresh medium containing 10% FBS in the absence or presence of amiloride (5 mM), sucrose (0.4 M), or nystatin (25 μg/mL), the cells were pre-incubated at 37 °C for 30 min. Then, the cells were treated with the relevant peptide (peptide concentration; 5 µM) and incubated for 30 min at 37 °C. For inhibition at 4 °C, the cells without the pre-incubation were treated with the relevant peptide (peptide concentration; 5 µM) and incubated for 30 min at 4 °C. After 30 min, the cells were washed with PBS containing heparin (20 units/mL) and added again, and detached by pipetting. The collected cells were washed once with PBS. Then, the cells were analyzed by a BD accuri C6 Plus flow cytometer (BD Bioscience, New Jersey). The inhibition rates were determined by normalizing the fluorescence intensity of cells incubated at 4 °C to that of cells incubated at 37 °C.

### 4.6. Preparation of Peptide/pDNA Complexes

Each 1 mM peptide solution in DMSO and pDNA was dissolved separately in 10 mM Hepes buffer (pH 7.3). To form peptide/pDNA complexes, peptide solutions of various concentrations were added to the pDNA solution in 2-fold excess. The final pDNA concentration was adjusted to 33.3 mg/mL. After the complex solutions were stored at room temperature for 15 min, the complex solutions were added. The charge ratio was defined as the residual molar ratio of the amino group of amino acid in the peptide to the phosphate group in the pDNA.

### 4.7. DLS Measurements and Zeta-Potential Measurements

The zeta-potentials of the peptide/pDNA complexes in 10 mM Hepes buffer (pH 7.3) were evaluated by the laser-Doppler electrophoresis method using Nano ZS (ZEN3600, Malvern Instruments, Ltd., Malvern, UK) with a He–Ne ion laser (633 nm). Light scattering data were obtained at a detection angle of 173° and temperature of 25 °C, and were subsequently analyzed by the cumulant method to obtain the hydrodynamic diameters, polydispersity indices (PDI) (m/G^2^) and zeta-potential of the complexes. Results were presented as the mean and standard deviation of 3 measurements (Table 2).

### 4.8. Cellular Uptake

HEK293 cells were seeded on 24-well culture plates (100,000 cells/well) and incubated overnight in 500 μL of DMEM. After exchange of medium, the peptide/Cy5 labeled pDNA complexes prepared at various charge ratios were applied to each well. The amount of Cy5-pDNA was 1 μg per well. After a 24 h incubation, medium was removed and cells were washed twice with PBS supplemented with heparin (20 units/mL), and detached by pipetting with PBS. The cells were centrifuged at 3000 rpm at 4 °C for 5 min. The cell pellets obtained were suspended in PBS and the cells were analyzed by a BD accuri C6 Plus flow cytometer (BD Bioscience, New Jersey).

### 4.9. Displacement Assay with Dextran Sulfate

Complex with 100 ng of pDNA and peptide at N/P ratio = 8 were incubated for 30 min at 37 °C in the presence of dextran sulfate (sodium salt, Wako, Japan) at various concentrations ranging 0, 0.05, 0.1, 0.5, and 1 mg/mL. After the incubation, the samples were analyzed on agarose gel and the gel was stained with SYBR Gold (Thermo Fisher Scientific, Waltham, MA, USA).

## Figures and Tables

**Figure 1 ijms-24-11768-f001:**
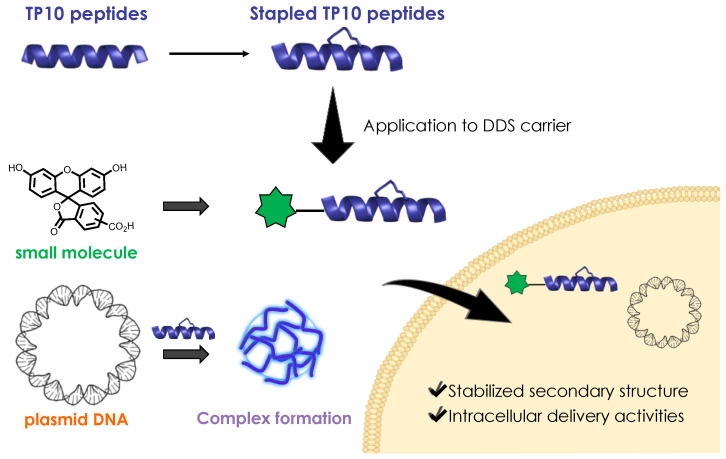
Concept of the delivery of small molecules and plasmid DNA using stapled hydrophobic cell-penetrating peptides.

**Figure 2 ijms-24-11768-f002:**
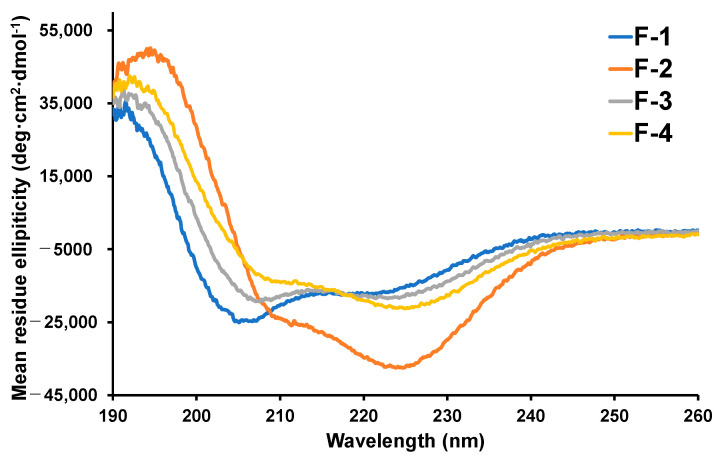
CD spectra of peptides **F-1**–**F-4** (10% acetonitrile in 20 mM phosphate buffer; peptide concentration: 50 µM).

**Figure 3 ijms-24-11768-f003:**
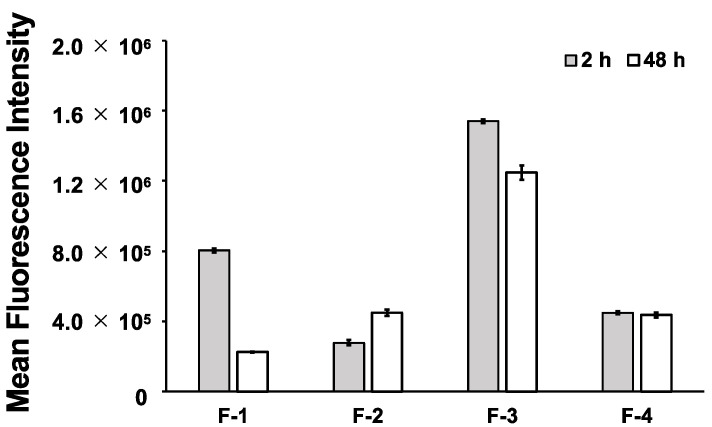
Evaluation of the intracellular delivery of 5(6)-carboxyfluorescein, as a model small molecule, by **F-1**–**F-4**.

**Figure 4 ijms-24-11768-f004:**
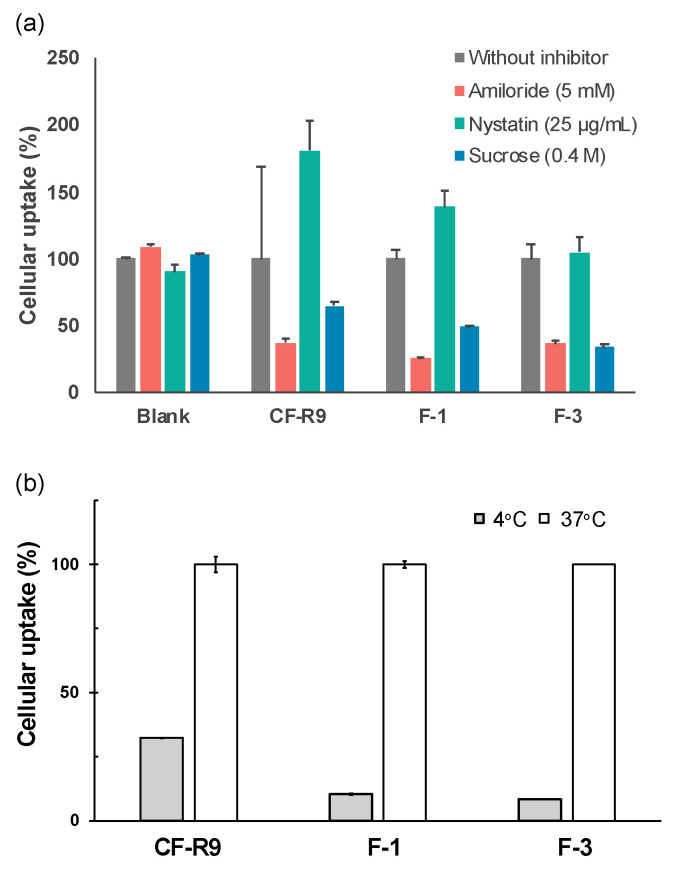
Evaluation of the cellular-uptake mechanism of **F-1** and **F-3**. (**a**) Using endocytosis inhibitors and (**b**) after incubation at 4 °C.

**Figure 5 ijms-24-11768-f005:**
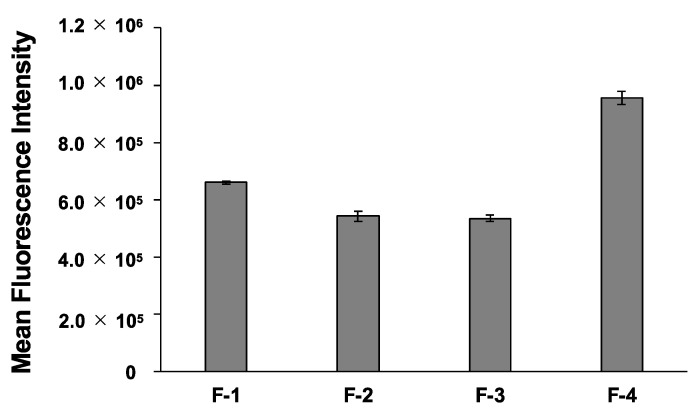
Cellular uptake of peptide/Cy5-pDNA complexes at N/P = 8 in HEK293 cells. The cellular uptake ability was determined by flow cytometry. The cells were excited with a 635 nm light for detecting Cy5-pDNA. Error bars represent the mean ± standard deviation, *n* = 4.

**Table 1 ijms-24-11768-t001:** Design of stapled TP10 analogs.

**Peptide**	**Sequence**	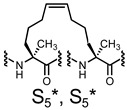
**TP10 (F-1)**	CF-XAGYLLGKINLKALAALAKKIL	
**F-2**	CF-XAS_5_*YLLS_5_*KINLKALAALAKKIL	
**F-3**	CF-XAGYLLGKINLKALAAS_5_*AKKS_5_*L	
**F-4**	CF-XAGYS_5_*LGKS_5_*NLKALAALAKKIL	

The *C*-terminus of each peptide had an amide (NH_2_) group. CF = 5(6)-carboxyfluorescein, X = β-alanine. The cyclization between *i* and *i* + 4 of the two (*S*)-4-pentenylalanine (S_5_) residues is indicated by an asterisk.

**Table 2 ijms-24-11768-t002:** Sizes, zeta-potentials, and polydispersity index (PDI) values of the peptide/pDNA complexes in solution prepared at charge ratios of 2, 4, and 8.

Peptide	N/P Ratio	Size (nm)	PDI	Zeta-Potential (mV)
**F-1**	2	3364 ± 689	0.707 ± 0.250	16.6 ± 1.2
	4	291 ± 40	0.368 ± 0.041	19.1 ± 1.9
	8	352 ± 71	0.407 ± 0.038	15.5 ± 2.1
**F-2**	2	226 ± 8	0.251 ± 0.018	23.4 ± 0.9
	4	291 ± 61	0.324 ± 0.097	23.4 ± 1.4
	8	410 ± 59	0.411 ± 0.058	16.7 ± 2.2
**F-3**	2	391 ± 8	0.417 ± 0.049	22.9 ± 1.1
	4	352 ± 57	0.366 ± 0.054	18.0 ± 1.7
	8	324 ± 53	0.399 ± 0.083	20.6 ± 3.0
**F-4**	2	480 ± 21	0.476 ± 0.015	19.3 ± 1.8
	4	263 ± 15	0.332 ± 0.032	15.6 ± 1.1
	8	392 ± 91	0.350 ± 0.042	16.7 ± 2.2

## Data Availability

The data presented in this study are available on request from the corresponding author.

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
