# Peer review of "Development of Hydrophobic Cell-Penetrating Stapled Peptides as Drug Carriers"

_ijms, 2023, doi:10.3390/ijms241411768_

Round 1

Reviewer 1 Report

The manuscript “development of hydrophobic cell-penetrating stapled peptides as drug carriers” by Yosuke Demizu et al reported TP10 engineering via peptide stapling. The stapled CPPs showed the improvement of alpha-helicity and cell permeability, which is via an endocytosis process. The stapled TP10 was conjugated with both a small molecular CF and pDNA as model study for cellular delivery. It is an interesting work and well-written with logic presentation, which would be good for a publication.

1.     F-2 showed the most stabilized peptide based on the CD measurement but why exhibited a poorer cell permeability compared to others?

2.     F-3 showed the most efficient CPP for deliver CF, while F-4 is the most efficient one for pDNA instead of F-3, any explanations about that?

3.     The efficiency of endosome escapes into cytoplasm of stapled CPPs may be good to be studied

none

Reviewer 2 Report

Review report

Journal: IJMS (ISSN 1422-0067)

Manuscript ID: ijms-2491543

Type: Article

Title: Development of hydrophobic cell-penetrating stapled peptides as drug carriers

Authors: Keisuke Tsuchiya, Kanako Horikoshi, Minami Fujita, Motoharu Hirano, Maho Miyamoto , Hidetomo Yokoo *, Yosuke Demizu *

Section: Molecular Biophysics

Special Issue: Novel Strategies in the Development of Peptide-Based Therapeutics

General

In this manuscript, the authors worked on the development of hydrophobic cell-penetrating peptides (CPPs) as drug carriers for the intracellular delivery of small molecules and nucleic acids. They synthesized the peptides using microwave-assisted solid-phase methods on an automated Liberty Blue™ microwave peptide synthesizer. To assess the delivery capacity of these TP10 derivatives, they selected CF [14] as a model small molecule and conjugated it to the N-terminus of each peptide. They further evaluated cellular absorption after treatment with endocytosis inhibitors and complete endocytosis inhibition at 4 °C. Notably, they observed that F-4 formed a complex with pDNA, which detached from the pDNA and migrated into the cells, while F-4 remained in the medium, contributing to maintaining the complex structure. The authors concluded that the TP10 derivatives developed in this study are promising peptides as delivery tools for small molecules and nucleic acids. They suggested that controlling the stapling position in hydrophobic CPPs could affect the delivery of different cargoes, a methodology expected to be useful in the development of hydrophobic CPPs as delivery tools.

Minor points

The authors have meticulously developed hydrophobic cell-penetrating peptides (CPPs) as potential drug carriers for the intracellular delivery of small molecules and nucleic acids. They have demonstrated this through a series of well-designed and executed experiments, providing a comprehensive evaluation of the delivery capacity of these peptides. Their findings offer promising implications for the use of these peptides in drug delivery, particularly with their suggestion that controlling the stapling position in hydrophobic CPPs could affect the delivery of different cargoes. The manuscript is well-written, providing clear and concise information, making it a valuable addition to the existing body of literature. However, these minor points should be clarified before its acceptance.

1.    I am particularly curious that the flow cytometry data provided do not come from density plots or scatter plots and only bar graphs are presented. The original raw data were not provided in the supplementary data. Please provide that information.

2.    It is suggested that the quality of the bar graphs in the manuscript be improved to align with the high standards of the International Journal of Molecular Sciences (IJMS). Enhancing the clarity, resolution, and overall presentation of these graphs will ensure that they effectively communicate the research findings and maintain the journal's reputation for publishing high-quality visual data.
